# Combining Mass Cytometry Data by CyTOFmerge Reveals Additional Cell Phenotypes in the Heterogeneous Ovarian Cancer Tumor Microenvironment: A Pilot Study

**DOI:** 10.3390/cancers15205106

**Published:** 2023-10-23

**Authors:** Liv Cecilie Vestrheim Thomsen, Katrin Kleinmanns, Shamundeeswari Anandan, Stein-Erik Gullaksen, Tamim Abdelaal, Grete Alrek Iversen, Lars Andreas Akslen, Emmet McCormack, Line Bjørge

**Affiliations:** 1Centre for Cancer Biomarkers CCBIO, Department of Clinical Science, University of Bergen, 5021 Bergen, Norway; 2Department of Obstetrics and Gynecology, Haukeland University Hospital, 5021 Bergen, Norway; 3Norwegian Institute of Public Health, 5015 Bergen, Norway; 4Delft Bioinformatics Laboratory, Delft University of Technology, 2628XE Delft, The Netherlands; 5Department of Radiology, Leiden University Medical Center, 2333ZA Leiden, The Netherlands; 6Centre for Cancer Biomarkers CCBIO, Department of Clinical Medicine, University of Bergen, 5021 Bergen, Norway; 7Department of Pathology, Haukeland University Hospital, 5021 Bergen, Norway; 8Centre for Pharmacy, Department of Clinical Science, University of Bergen, 5021 Bergen, Norway

**Keywords:** high-grade serous ovarian cancer (HGSOC), mass cytometry, tumor microenvironment (TME), immunophenotyping, heterogeneity, merging algorithm

## Abstract

**Simple Summary:**

High-grade serous ovarian cancer (HGSOC) has a dismal prognosis, but its tumor microenvironment (TME), which impacts disease progression and prognosis, is inadequately mapped. A better understanding of the complexity of the TME requires its characterization with multidimensional approaches that allow for simultaneous identification and categorization of the various cell populations. Here, we have performed in-depth profiling of the cellular TME by merging two high-dimensional single-cell datasets generated by mass cytometry of dissociated tumors, to determine whether combining datasets by the CyTOFmerge algorithm can provide more information on the HGSOC TME than separate dataset analyses. The results confirmed high interpatient heterogeneity and that such merging had the potential for identifying novel combinations of markers expressed on cells of the TME. These hypothesis-generating findings could help identify new combinations of expressed antigens and lead to improved mapping of the HGSOC TME and enhanced response to therapy.

**Abstract:**

The prognosis of high-grade serous ovarian carcinoma (HGSOC) is poor, and treatment selection is challenging. A heterogeneous tumor microenvironment (TME) characterizes HGSOC and influences tumor growth, progression, and therapy response. Better characterization with multidimensional approaches for simultaneous identification and categorization of the various cell populations is needed to map the TME complexity. While mass cytometry allows the simultaneous detection of around 40 proteins, the CyTOFmerge MATLAB algorithm integrates data sets and extends the phenotyping. This pilot study explored the potential of combining two datasets for improved TME phenotyping by profiling single-cell suspensions from ten chemo-naïve HGSOC tumors by mass cytometry. A 35-marker pan-tumor dataset and a 34-marker pan-immune dataset were analyzed separately and combined with the CyTOFmerge, merging 18 shared markers. While the merged analysis confirmed heterogeneity across patients, it also identified a main tumor cell subset, additionally to the nine identified by the pan-tumor panel. Furthermore, the expression of traditional immune cell markers on tumor and stromal cells was revealed, as were marker combinations that have rarely been examined on individual cells. This study demonstrates the potential of merging mass cytometry data to generate new hypotheses on tumor biology and predictive biomarker research in HGSOC that could improve treatment effectiveness.

## 1. Introduction

High-grade serous ovarian cancer (HGSOC) is the most common subtype of ovarian cancer and is also the leading cause of death among women with epithelial ovarian cancer (EOC) [1,2,3], with the long-term survival rates only having improved modestly over the last few decades. HGSOC is characterized by late-stage diagnosis and acquired chemoresistance. The introduction of poly-ADP-ribose polymerase inhibitors as part of frontline therapy has seemingly translated into an overall survival benefit, at least for a subgroup of patients with homologous recombination deficiency or mutated *BRCA* genes. Unfortunately, a large proportion of HGSOC patients does not fall into these categories.

The profiling of molecular patterns, such as *BRCA* mutations and homologous recombination defects, and phenotypic traits, such as platinum sensitivity and degree of surgical debulking, and the integration of this information into clinical trials and wider practice has improved our understanding of the pathogenesis of HGSOC. The molecular landscape of HGSOC is characterized by inherent inter- and intra-tumoral heterogeneity, which results in a spectrum of disease phenotypes. This heterogeneity is found on a genetic, immunologic, and cellular level. Cancer research has moved from primarily focusing on tumor cells to investigating these cells as part of the tumor microenvironment (TME). The HGSOC TME encompasses several cell types in addition to cancer cells, such as infiltrating immune cells, and stromal cells like fibroblasts, as well as non-cellular components such as extracellular matrix proteins, and secreted molecules [4,5,6,7]. The cells in the TME are dynamic and exhibit plasticity; this feature allows them to switch between phenotypes and contribute to the reciprocal relationship between the TME components found across tumor types, and which seems pivotal for tumor growth, progression, and response to therapy [5,7,8,9].

The tumor immune microenvironment (TiME) varies substantially with regard to the degree of effector T-cell infiltration of the epithelium and the presence of pro-inflammatory cytokines [10,11,12,13]. EOC was one of the first cancers in which increased density of tumor-infiltrating lymphocytes and the presence of regulatory T cell (Treg)-mediated immunosuppression in the TiME were found to have a survival benefit [14,15,16]. Another feature of the TiME of EOC is that the anti-inflammatory M2-type macrophages are overrepresented. This macrophage phenotype has been associated with immune evasion and tumor growth, and accordingly, a suppressive TiME is associated with highly evolved immune escape mechanisms and worse outcomes [17,18]. This immunosuppression feature, which is common in EOC, might partly explain both why immunotherapy, which has revolutionized treatment outcomes for many solid cancers [4,8], has shown marginal efficacy in HGSOC, and why the biomarkers used to identify responders in other tumors might not be clinically relevant in this disease.

The ideas and methods for the profiling of tumors are constantly evolving. While histopathological and immunohistochemistry methods form the basis for diagnosis in clinical practice, genomic sequencing methods are increasingly being used. In addition, the profiling of single cells (including sequencing based on DNA and RNA, protein, or immune cell profiling) and evaluation of multiple tumor sites have shown promise for providing better insights into tumor biology as well as predicting metastatic potential and, thus, aiding in drug development efforts [19,20]. Combining these methods may enable a higher degree of molecular mapping of the biological variability of tumors and provide better support for clinical decision-making. Furthermore, different forms of profiling are required to test distinct research hypotheses. Most of the molecular research performed to date is genetically based. According to the results of such genetics-based molecular methods, HGSOC is characterized by genomic structural variations, such as copy number aberrations, that cause changes in global gene expression. The resulting range of chromosomally unstable tumors has been partly classified by copy number signatures, gene expression profiles, and their ability for DNA repair through homologous recombination. The methods applied in such research are based on a statistical examination of the genomic data map rather than an illustration of the dynamic real-time changes in the TME. In contrast, spatial resolution and interrogation of cell–cell interactions could enable a real-time understanding of tumor biology. 

The mass cytometry, or cytometry by time-of-flight (CyTOF), method is useful for the identification of single-cell phenotypes and interpatient tumor heterogeneity. The technique utilizes metal isotope-conjugated antibodies to detect the cellular expression of antigens: more than 40 parameters per cell can be profiled at the single-cell level with this method [21]. Although the body of studies profiling HGSOC tissues by this method is on the rise, so far, the research on this topic has mainly focused on either immune or tumor cells, rather than the spectrum of cells constituting the TME. Previously, studies have demonstrated how mass cytometry could be applied to identify interpatient TME heterogeneity [22,23,24,25,26]. For example, Gonzalez et al. applied the method to 17 single-site HGSOC samples and identified several tumor cell populations co-occurring in the cohort, including two cell subsets with low frequency that were correlated with worse outcomes [23]. The same samples and method have been used to discover an NK cell subset associated with tumor cell abundance [24].

The development of merging algorithms has enabled the combination of several datasets for improved data exploration, and the approach has also been adopted for mass cytometry. CyTOFmerge is a MATLAB algorithm for the merging of two mass cytometry datasets based on overlapping markers and imputation by a k-nearest-neighbor algorithm of those markers missing from each mass cytometry panel [27]. With this method, a wider set of cellular antibody markers and, thereby, tumor cells and immune cells of myeloid and lymphoid lineages can be explored concurrently. Such combined methods may be required to identify the clinical effects of treatment, including the array of immunotherapeutic modalities available currently.

This pilot study explored the potential of combining datasets assembled through two mass cytometry panels for better coverage of the cell subsets constituting the HGSOC TME. After an immune-focused panel and a pan-tumor panel of antibodies were applied to single-cell suspensions from a small but well-characterized HGSOC cohort, the generated datasets were merged by the CyTOFmerge MATLAB algorithm. This method led to the identification of an additional tumor cell subset, other than the nine detected by the pan-tumor panel alone. Furthermore, this study demonstrated how merging and data imputation could help identify the cellular co-expression of markers that are usually not examined together, such as immune cell markers found on tumor and stromal cells. A few possible associations of these cell populations with clinical outcome parameters were also identified. To further validate and investigate these findings, a mass cytometry panel combining the markers of interest identified in this study, and a larger cohort is required, in addition to other single-cell profiling methods and biopsies from multiple sites.

## 2. Materials and Methods

### 2.1. Patient Cohort

This study used tumor tissues from ten patients diagnosed with HGSOC at the Department of Obstetrics and Gynecology, Haukeland University Hospital, Bergen, Norway, whose samples were deposited over a two-year period (2016–2018) in the Bergen Gynecologic Cancer Biobank.

Based on the inclusion criteria, the study included chemo-naïve patients who were newly diagnosed with ovarian cancer and were scheduled to undergo primary cytoreductive surgery, with or without adjuvant chemotherapy, as their primary treatment. All Federation Internationale de Gynecolgie et d’Obstetrique (FIGO) stages could be included (Table 1). For all the participants, the diagnosis of HGSOC was verified by gyne-pathologists postoperatively, as part of the clinical routine.

Progression-free survival (PFS) was defined as the interval from the end of primary treatment to the identification of recurrence, while overall survival (OS) was measured from the end of primary treatment to the time of death or the database cut-off date (which was 22 October 2020). The level of tumor debulking was classified according to the remnant visible tumor tissues in the intraabdominal region at the end of the surgical procedure (R < 1 cm, or R > 1 cm). Clinical status was noted according to whether the patient was alive (with or without cancer recurrence) or dead (either from cancer or from other causes) at the data cut-off time point.

### 2.2. Sample Processing

#### 2.2.1. Tumor Tissues

During the primary cancer procedure, 1–2 cm^3^ of representative tumor tissue was sampled from each patient, directly after the primary tumor had been removed from the abdominal cavity. The tumor tissue samples were promptly enzymatically dissociated to obtain single-cell suspensions (Appendix A), and cell viability was determined by trypan blue staining. Cells from all the collected samples were cryopreserved at −150 °C in a freezing medium that consisted of 90% fetal calf serum (FCS) (Gibco, Thermo Fisher Scientific, Waltham, MA, USA) and 10% dimethyl sulfoxide (Sigma Aldrich, Merck KGaA, Darmstadt, Germany).

#### 2.2.2. Quality Control Samples

To ensure that the markers would indicate cell expression, if present, peripheral blood mononuclear cells (PBMCs) and the human ovarian serous adenocarcinoma (OV-90) cell line were included as quality controls.

PBMCs were isolated from whole blood samples collected from healthy donors at the Blood Bank, Department of Immunology and Transfusion Medicine, Haukeland University Hospital, Bergen, Norway. The isolation of PBMCs was performed by density-gradient centrifugation with Lymphoprep (Axis-Shield, Oslo, Norway), after which the cells were cryopreserved at −80 °C in 90% heat-inactivated FCS (Gibco, Thermo Fisher Scientific, Waltham, MA, USA) and 10% dimethyl sulfoxide (Sigma Aldrich, Merck KGaA, Darmstadt, Germany).

OV-90 cells (American Type Culture Collection [ATCC]^®^CRL-11732™) were cultivated and expanded in RPMI 1640 medium supplemented with 100 IU/mL penicillin (Gibco, Thermo Fisher Scientific, Waltham, MA, USA), 10% heat-inactivated FCS, and 2 mM l-glutamine, and were then cryopreserved at −80 °C. 

The presence of immune cells in the dissociated tumor tissues was confirmed in adjacent and representative slides of paraffin-embedded tissues from the same primary tumors. The tissue slides were stained with eosin-hematoxylin and an anti-CD45 antibody (Ventana Bench-Mark Ultra Platform; Roche, F. Hoffmann-La Roche Ltd., Basel, Switzerland), and binding of the antibody to the target antigen was detected by the UltraView system (Ventana BenchMark Ultra platform, Roche, F. Hoffmann-La Roche Ltd., Basel, Switzerland) (Appendix A).

### 2.3. Ethical Statement

The Regional Committees for Medical Research Ethics have approved the Bergen Gynecologic Cancer Biobank (ID2014/1907) and the healthy donor biobank (ID2212/2247), as well as provided approval for the present study (ID2017/623). The informed consent of all the participants has been obtained. 

### 2.4. Antibodies for Mass Cytometry

#### 2.4.1. The Pan-Tumor Panel

To investigate all the main cell subsets of the TME, a panel containing 35 antibodies was used (Appendix A) [22]. The panel included markers for the detection of tumor cells (n = 11), stromal cells (n = 4), and immune cells (n = 15). Additionally, the panel contained a few, select, immune checkpoint markers (n = 6). The antibodies were either obtained in the pre-conjugated state from the manufacturer (Fluidigm, San Francisco, CA, USA) (n = 14) or conjugated in-house (n = 11). 

#### 2.4.2. The Pan-Immune Panel

To characterize the immune cell subsets in dissociated tumors, an established panel of 34 antibodies was used (Appendix A) [28]. In addition to markers for identifying tumor cells and stromal cells (for their exclusion), the panel encompassed markers for the detection of naïve and activated T cells, antigen-presenting cells, NK cells, B cells, and macrophages. Metal-conjugated antibodies (n = 28) and purified monoclonal antibodies (n = 6) were purchased from Fluidigm (San Francisco, CA, USA), BioLegend (San Diego, CA, USA), and Abcam (Cambridge, MA, USA).

#### 2.4.3. Antibody CONJUGATION and Validation

All the in-house conjugated monoclonal antibodies were conjugated to pre-determined metal isotopes with the MaxPar Antibody Labeling Kit X8 4Rxn (Fluidigm, San Francisco, CA, USA) according to the producer’s protocol. Directly after this step, the conjugated antibodies were diluted to 1 µg/µL in Ab Stabilizer Solution (Candor Biosciences, Wangen, Germany) supplemented with 0.01% sodium azide (Santa Cruz Biotechnology, Dallas, TX, USA) and stored at 4 °C. 

The optimal concentrations of all the conjugated antibodies, including the pre-conjugated ones, were determined by titration of serial dilutions on dissociated ovarian tumor tissues, PBMCs, and OV-90 cells.

### 2.5. Mass Cytometry Analysis

For the mass cytometry analyses, the dissociated tumor tissue samples and validation samples were thawed to room temperature and stained with 5 µM of Cell-ID^TM^ Cisplatin (Fluidigm, CA, USA), following which fixation and pre-permeabilization palladium barcoding were performed as described previously [22]. A barcoded sample mixture pool containing the ten patient samples, the healthy donor PBMCs, and OV-90 cells was prepared. The barcoded sample mixture pool was divided into two equal portions. Both barcoded sample mixtures were washed twice with 2 mL of cell-staining buffer (CSB, Fluidigm, San Francisco, CA, USA) and centrifuged at 800× *g* for 4 min at 4 °C. Next, the following steps were performed at room temperature. The samples were incubated with FcR blocking reagent (Miltenyi Biotec, Bergisch Gladbach, Germany) at a 1:10 dilution for 20 min prior to intra- and extracellular staining. The barcoded sample mixtures were stained with the relevant extracellular antibody cocktail (1:1 sample-to-antibody cocktail volume ratio) for 20 min: the pan-tumor panel antibodies were applied to one portion of the mixture, and the pan-immune panel antibodies were applied to the other portion, according to predetermined dilutions (Appendix A). Thereafter, the pan-tumor sample mixture was washed twice at 800× *g* with CSB, and 1 mL of methanol was added. Following this, the intracellular marker stain (1:1 sample-to-antibody cocktail volume ratio) was added for 20 min. After the last antibody staining step for each of the panels, the stained cells were washed twice with CSB and resuspended in 1 mL of nucleic acid Ir-Intercalator (Fluidigm, San Francisco, CA, USA) for 1 h. The samples were then suspended in CSB and centrifuged at 800× *g* for 4 min. The pellets obtained were suspended in Maxpar PBS and centrifuged again at 800× *g* for 4 min. The pellets containing the samples were then diluted in Cell Acquisition Solution with 10% EQ Four Element Calibration Beads to a final concentration of 1 × 10^6^ cells/mL (all the reagents were purchased from Fluidigm, San Francisco, CA, USA). The samples containing the samples were injected through a wide-bore injector system (Fluidigm, San Francisco, CA, USA) and analyzed on a Helios mass cytometer with the Helios 6.5.358 acquisition software (Fluidigm, San Francisco, CA, USA).

### 2.6. Mass Cytometry Data Analysis

The acquired mass cytometry data were normalized using internal bead standards and debarcoded with a 20-plex-debarcoding key (CyTOF software, v6.7, Fluidigm, San Francisco, CA, USA). Initial data cleaning of the debarcoded files, including gating for live cells and then for single cells, was performed on the Cytobank analysis platform (v7.3.0; Beckman Coulter Inc., Indianapolis, IN, USA) (Appendix A). As a quality control step, Uniform Manifold Approximation and Projection for Dimension Reduction (UMAP) plots of all the samples, including the two control samples, were created to assess the phenotyping ability of the panel antibodies and the distribution of immune cells across tissues when examined by the two mass cytometry panels (Appendix A). The two panels had a total of 35 (pan-tumor) and 34 (pan-immune) markers, of which 18 were overlapping. Thus, one of the panels contained 17 unique markers and the other one contained 16 unique markers. 

To examine if the data depth increased when the two datasets were combined, the pan-tumor dataset and the pan-immune dataset were combined using CyTOFmerge, an established merging algorithm [27]. In short, based on the set of overlapping markers in the two panels, markers that were not measured in one panel were imputed according to the measurements from the complementary panel. The set of markers that best preserved the high-dimensional structure of the data was determined by a dimensionality reduction technique (principal component analysis). For both panels, data on which cell populations are present, based on a comparable expression of protein markers, were imputed using the k-nearest-neighbor algorithm and clustered by Phenograph in the CyTOFmerge algorithm [29].

After the merging procedure had been performed, the mass cytometry files from the separate pan-tumor and pan-immune panel runs and the file containing the merged data were uploaded and analyzed with the Astrolabe Cytometry Platform (Astrolabe Diagnostics Inc., Arlington, VA, USA). The Astrolabe platform is a fully automated analysis pipeline that labels each sample separately based on the Ek’Balam algorithm [30]) of pre-determined cell subsets (Appendix A), according to the definitions outlined by Maecker et al. [31] and Finak et al. [32]. Further data clustering was performed using the consensus hierarchical clustering tool FlowSOM (R package FlowSOM 2.8.0 [33]), and Pearson’s correlation was applied as a distance metric in the hierarchical clustering for heatmaps. 

The first clustering of the cells in the pan-tumor panel dataset and the merged panels dataset was examined based on a custom-made clustering definition (Appendix A). The immune cells (CD45^+^cells) were named according to the Astrolabe standard naming algorithm. The non-immune cell populations were defined as CD45^−^cells and thereafter labeled as CD56^+^(mainly fibroblasts) or CD56^−^cells (tumor cells). The tumor cells were further stratified according to the expression of epithelial cell adhesion molecule (EpCAM), folate receptor 1 (FOLR1), and CD24. In contrast, the pan-immune panel was analyzed according to the standard Astrolabe pre-determined cell subset clustering of CD45^+^cells (Appendix A and Appendix A), with the CD45^−^cells excluded from further analyses.

### 2.7. Statistical Analysis

The initial quality control and primary gating steps for removing cell doublets and debris and gating out live single cells, as well as visualizations by Uniform Manifold Approximation and Projection (UMAP) plots, were performed with the Cytobank analysis software (v8.0 and v9.1; Beckman Coulter Inc., Indianapolis, IN, USA). In addition to the analysis tools embedded in the Astrolabe platform, statistical analysis was performed using Microsoft Excel v16.78 (Redmond, WA, USA), GraphPad Prism 9 (San Diego, CA, USA), and R [34].

The visualization of the results and differential abundance analyses were performed with the R package edgeRv3.11 and GraphPad Prism 9, and only cell subsets with cell counts ≥5 were included in the primary results. Both the pre-defined immune cell populations and the cell subsets identified by the Astrolabe platform through the unsupervised clustering procedure FlowSOM were used to determine potential clinically relevant differences between samples. Paired t-testing was performed for comparisons of two groups, as indicated in the figure legends. Data are expressed as arithmetic mean ± standard deviation (SD). The statistical tests used and the statistically significant findings for each experiment are shown in the corresponding figure legends.

Adjusted *p* values < 0.01 were considered statistically significant. Baseline values for the differential abundance analyses were as follows: debulking status = R0, no recurrence; clinical status = alive without disease.

## 3. Results

### 3.1. Patient Characteristics

The patient cohort included ten chemo-naïve patients with HGSOC (Table 1). Seven of the patients had no macroscopically visible residual disease by the end of the primary cytoreductive surgical procedure (R0), while in three others, complete cytoreduction was not achieved (R > 1, patients 3, 4, and 9). At the database cut-off time point, eight patients were alive and the median overall survival of the cohort was 14.6 months (Appendix A). Of the ten patients, the disease had recurred in four (Table 1, Appendix A).

### 3.2. Characterization of the Ovarian TME with the Pan-Tumor Panel

First, we aimed to identify cell phenotypes within the tumor samples of all ten patients using the pan-tumor panel to investigate tumor cell heterogeneity and the extent of inter-patient heterogeneity. The cell distribution of the examined tumors demonstrated heterogeneity across patients, with tumor cells comprising the main bulk of the identified 21 different cell phenotypes found in most of the individual tumor samples, except for the samples from P2 and P5 (median, 71.1%; range, 32.6–91.3%). In contrast, the median fibroblast frequency was 11.1% (range, 2.0–27.2%), and the immune cell prevalence ranged between 4.2% and 59.8% (median, 12.3%) (Appendix A). A small subset of the total cells (median, 0.1%; range, 0.02–0.9%) could not be categorized into either of these three populations (tumor cells, fibroblasts, immune cells), but was classified as debris based on their expression of markers and excluded from further analyses.

While the tumors of three patients (P2, P4, and P7) had higher B cell (1%, 3%, and 1%, respectively) and CD4^+^T cell (4%, 1%, and 2%, respectively) fractions, the frequencies of these subsets in the remaining eight tumors ranged from 0.02% to 0.1%. Furthermore, natural killer (NK) cells were missing in 3 of the 10 tumors (P1, P3, and P7), and natural killer T (NKT) cells were not present in 4 of the 10 tumors (P2, P3, P6, and P7). The composition of the different tumor cell subtypes varied across the tumor samples. Patients P1, P2, P3, and P4 lacked EpCAM^−^FOLR1^+^CD24^−^cells. Three of these patients (P1, P3, and P4) lacked the EpCAM^−^FOLR1^+^CD24^+^cell subset. Three (P2, P3, and P4) lacked the triple-negative cell subset, while P1 demonstrated the highest proportion of the triple-negative cell subset (43%). Furthermore, P3 and P4 had the highest proportion of EpCAM^+^FOLR1^+^CD24^−^cells (38% and 33%, respectively), and also had among the largest fractions of triple-positive cells (11% and 12%, respectively). With regard to the stromal cells, all the tumors contained the CD56^+^fibroblast subset at a frequency ranging from 4% to 27% (Appendix A).

CyTOF analysis of the pan-tumor dataset revealed ten tumor/stromal, nine immune, and two undefined cell populations (CM cells^−^, Figure 1A). The tumor/stromal cells included one CD56^+^fibroblast population and nine CD45^−^CD56^−^cell populations, which were distinguished by the expression of the three select tumor markers, namely, EpCAM, FOLR1, and CD24. Two of these tumor cell populations expressed EpCAM, FOLR1, and CD24 simultaneously (Figure 1A). One of the tumor cell populations, CD45^−^CD56^−^(EpCAM^+^FOLR1^+^CD24^+^), demonstrated high expression of the markers. The cells of the other tumor cell population, CD45^−^CD56^−^unassigned, expressed the three markers, but with medium intensity. The identified immune cells included five T cells, one B cell, and one NK cell population of the lymphoid lineage. From the myeloid lineage, one monocyte/macrophage and one mast cell population were detected. Additionally, the algorithm identified two CD45^+^CD11b^+^immune cell populations that differed in terms of HLA-DR, CD44, and CD4 expression: CM^−^HLA-DR^+^CD44^dim^CD4^dim^unassigned cells and CM^−^HLA-DR^−^CD44^−^CD4^−^unassigned cells.

### 3.3. Characterization of the Ovarian TiME with the Pan-Immune Panel

Next, analysis of the CD45^+^cells based on the pan-immune panel revealed the presence of immune cells of both the myeloid and the lymphoid phenotypes. Both mass cytometry analysis of tumor cell suspensions and solid tumor immunohistochemistry analyses of CD45^+^cell frequencies revealed interpatient differences (Appendix A and Appendix A). The identified CD45^+^cells comprised 5.4% to 82.4% of all single cells in the pan-immune dataset. When tissue cell suspensions were assessed and the immune cells were divided into the main categories (that is, B cells, granulocytes, T cells, NK cells, and myeloid cells), a high abundance of “other” cells was revealed (Appendix A). These “other” cells included lineage (lin)-negative cells, in addition to debris and cell doublets (Appendix A). One of the lin^−^cell populations was HLA-DR^high^CD33^low^, which also expressed the myeloid markers CD11c, CD141, CD162, and CD86 (marker expression intensity > 1).

Based on the downstream analysis, the CD45^+^cells were divided into 29 main immune cell populations (Figure 1B and Appendix A). One of these computationally assigned populations comprised cell doublets and debris and was excluded from further analysis. Among the remaining 28 analyzed immune cell populations (Figure 1B), a total of 16 lymphoid cell populations were identified, including B cells (n = 2), NK cells (n = 3), CD4^+^T cells (n = 5), CD8^+^T cells (n = 4), double-negative T cells (n = 1), and double-positive T cells (n = 1) (Appendix A). Additionally, nine myeloid populations that included four different dendritic cell (DC) subtypes, four monocyte/macrophage populations, and one mast cell population were identified. The presence of these cell populations varied across samples. While granulocytes (P2, P6, and P10) and B cells (P2, P4, P6, P8, and P9) were detected only in some samples, most samples contained T cells, NK cells, and myeloid cells at higher frequencies (Appendix A). NKT cells (CD56^dim^NK cells expressing CD3) were not among the frequently detected cell populations in any of the patients. P9 and P10, who were alive but had disease relapse, displayed the highest frequencies of T cells (32.5% and 17.5%, respectively). The highest density of CD8^+^T cells was detected in P9 (5.3%), which was followed by P2 (3.6%), P10 (1.5%), P4 (1.3%), and P1 (1.1%), in descending order. In the remaining five patient samples, the frequency of CD8^+^T cells was below 1%.

Of the 28 main immune cell populations, the following seven cell populations were detected with the highest frequency across all the patient samples: (1) double-negative T cells (4.8 ± 3.1%), (2) conventional CD1c^−^CD141^−^DCs (16.3 ± 15.3%), (3) type 1 CD141^+^DCs (4.7 ± 3.7%), (4) CD14^+^CD16^−^classical monocytes (or macrophages) (2.5 ± 2.1%), (5) CD56^+^CD16^−^NK cells (14.5 ± 12.6%), (6) hematopoietic lin^−^HLA-DR^+^cells (26.2 ± 12.5%), and (7) lin^−^/unassigned cells (21.8 ± 14.8%) (Appendix A). Extended phenotypic characterization of the lin-negative/unassigned cell population (number 7) demonstrated a range of activation markers that were expressed (CD45RO, CD62L, CD86, CD95, and CCR7) (Appendix A). Additionally, the most frequent cell population, which was hematopoietic lin^−^HLA-DR^+^ cells, expressed CD11c, CD33, CD141, and CD163, which could indicate that it was a tumor-infiltrating macrophage population (Appendix A).

### 3.4. TME Information Generated by Merging the Pan-Tumor and Pan-Immune Data

Then the two data sets were combined by the analysis algorithm (CyTOFmerge MATLAB) to investigate if such merging could reveal additional cell phenotypes, extending the understanding of the ovarian TME. The merged dataset was first divided into a total of 40 cell populations, including 29 immune, 9 tumor, and 1 stromal cell population, as well as 1 lineage-negative population, according to the pre-defined labeling hierarchy (Figure 1C and Appendix A). The frequency of each population varied between cell populations and across samples (Figure 2, Appendix A). The nine tumor cell populations in the merged dataset defined by the Astrolabe algorithm matched the nine tumor cell populations identified in the pan-tumor dataset (Figure 1, Addendum 1). Among the 29 immune cell clusters, we identified a total of 17 lymphoid cell subsets, including 11 T cell, 3 B cell, and 3 NK cell populations, with the last one including one group of CD56^dim^CD3^+^NKT cells. From the myeloid lineage, four DCs, three monocyte/macrophages, one granulocyte, one mast cell, and three unclassified myeloid cell subsets were identified. When the data was subdivided further by hierarchical clustering, a total of 206 cell subsets were identified, among which 199 were lineage-positive. Of these, 156 were immune cell subsets, while 39 tumor cell subsets and 5 stromal/fibroblast subsets made up the remaining populations. In comparison, hierarchical clustering of the pan-tumor and pan-immune datasets demonstrated 114 and 177 lin-positive cell subsets, respectively.

The median antibody expression in the pre-determined 39 linage-positive cell populations was compared between the merged data and the pan-tumor or pan-immune panel data. Cell populations that were identified across all three datasets were found to be present in the same samples across the datasets. The cell subsets found in all three datasets included double-negative T cells, double-positive T cells, mast cells, and NKT cells (Addendum 1). The expression intensity of the markers was similar for the merged data and the original dataset for the cell populations identified with either the pan-tumor panel (fibroblasts and EpCAM^+^FOLR1^+^CD24^+^tumor cells) or the pan-immune panel (Tregs and CD8^+^T cells [EMRA: terminally differentiated effector memory cells re-expressing CD45RA]) (Addendum 1). In a few samples, there was a mismatch in the cell expression of antibodies between the original (either pan-tumor or pan-immune) data and the merged dataset. In those cases, some of the antibodies expressed by defined cell populations in the original datasets could not be identified for the same population in the merged dataset. This phenomenon was essentially seen when the cell populations identified by the pan-tumor/pan-immune panels demonstrated low expression of those particular markers.

#### 3.4.1. Novel Cell Subsets 

Further examination of the 39 populations detected by the merging algorithm demonstrated that combining the panels led to an increase in the number of expressed markers on the cell populations (Figure 1C). Since the CD45-cells were excluded from the pan-immune data in the Astrolabe analyses, comparisons of non-immune cell subsets in the Astrolabe results generated from the pan-immune data and from the other data sets were not feasible. In the triple-positive tumor cells (EpCAM^+^FOLR1^+^CD24^+^), the merging of the two datasets resulted in the expression of additional markers traditionally not examined in tumor cells: CD33, CD303, and CD95, which are involved in cell signaling, antigen capture, and immune homeostasis and cell apoptosis, respectively (Addendum 1).

The algorithm incorporated in the Astrolabe platform identified cell subsets containing >5 cells. To further decrease the number of false-positive cell subsets, in this study only hierarchically clustered cell subsets composed of at least 50 cells across the samples in a dataset were compared between the pan-immune (n = 141), pan-tumor (n = 106), and merged (n = 163) datasets. The clustering algorithm in Astrolabe automatically generated subset names based on markers highly expressed on the cells in the subset. This prioritization of markers differed between the three data sets, resulting in a discrepancy in the subset names between the datasets. While naming was matched for several cell subsets, for some cell subsets it differed between the pan-cancer/pan-immune data and the merged data. This partial mismatch in marker expression posed a challenge for further analyses, but one subset, the EpCAM^−^FOLR1^−^CD24^+^TAG72^lo^CD11b^lo^CCR7^lo^ tumor cells, was identified with certainty as being novel in the merged dataset. These CD24^+^cells were CD47^+^ and further demonstrated weak expression of CD133, CD33, and CD95.

From the smaller pan-tumor panel, only one fibroblast population was found to match the merged dataset based on the expression intensity of the pre-defined markers. In contrast, five fibroblast subsets were found to be matched between the pan-immune and the merged datasets. While the algorithm identified the fibroblasts according to the expression of EpCAM in all datasets, further clustering was performed based on the estimated expression of FOLR1 and aSMA in the pan-tumor and merged datasets (Figure 3A,C and Appendix A) and CD73 expression in the merged dataset (Figure 3B,D and Appendix A). All the fibroblast subsets expressed high levels of CD24, CD47, CD95 (imputed), and CD33 (imputed), but in the merged dataset, the different fibroblast subsets demonstrated differences in the expression of the following markers: CD133, CD45RO, FAPa, and PDGFRb, as well as the imputed markers CCR7, CD123, CD141, CD27, and CD62L (Figure 3C,D).

#### 3.4.2. Heterogeneity in the Cell Subset Composition of Tumor Samples

As expected, the tumors displayed a heterogeneous cell composition (Figure 4, Addendum 1). While 14 of the cell populations could be identified in all the samples in varying proportions, including double-negative T cells, all CD4^+^T cells, naive CD8^+^T cells, effector memory CD8^+^cells, CD8^+^cells (terminally differentiated effector memory cells re-expressing CD45RA; EMRA), fibroblasts, and triple-negative (EpCAM^−^FOLR1^−^CD24^−^)CD45^−^CD56^−^cells, the remaining 25 populations were only found in some of the tumor samples (Addendum 1). The diverging cell populations included CD1c^−^CD141^−^DCs (median = 3132, range = 0–80,848), double-positive T cells (median = 9, range = 0–91), and unassigned (HLA-DR^+^/^−^)cells (median = 2640 and 1344; range, 0–34,238 and 0–363,385, respectively), as well as most of the identified tumor cell populations. When conventional CD1c^−^CD141^−^DCs and the HLA^−^DR^−^unassigned cells were excluded, a matching heterogeneous pattern in the 25 populations could be seen between samples in the merged dataset and the pan-tumor and/or the pan-immune datasets.

After the merged data had been reduced to cell subsets of at least 50 cells, it contained a total of 141 lineage-positive cell subsets (Figure 4). The 22 lineage-negative cell subsets were excluded from further analyses. When markers from the two datasets were combined, potential differences between patients emerged. In this study, differences were observed in PD1 expression on effector memory CTLA4^low^CD103^low^CD8^+^T cells, in CD163 and CD25 expression on type 1 CD141^+^CD123^lo^DCs, and in a range of markers expressed on monocytes/macrophages and tumor cells (Figure 4).

### 3.5. Association of the Detected Cell Subsets with Clinical Parameters

When the clinical features of the ten patients were analyzed against the main cell subsets in each of the datasets, several significant associations were revealed (*p* < 0.01) (Table 2). From the pan-tumor dataset, one cell population was associated with disease recurrence, and from the pan-immune dataset, two cell populations were associated with clinical status. From the merged dataset, seven associations were identified (Table 2). When the subdivided cell populations were examined against three clinical phenotypes (surgical debulking status, disease recurrence, and clinical status), associations were detected for 3, 15, and 30 cell subsets in the pan-tumor, pan-immune, and merged datasets, respectively (Appendix A). 

## 4. Discussion

HGSOC is a disease characterized by biological and phenotypic heterogeneity. Despite extensive research efforts, few established treatment-relevant biomarkers exist, other than the *BRCA1/2* mutation status and the capacity for homologous recombination repair. This is partly attributable to the methodological challenges of conducting a thorough examination spanning all the cell types contained within the TME. The ability of mass cytometry to uncover the full heterogeneity of a tumor sample is, at present, restricted by a lack of a sufficient number of markers within a single panel. This pilot study demonstrates the potential of bioinformatically merging two mass cytometry datasets from the same patient cohort to dive further into the phenotypic diversity of the TME. While the merging of the two datasets identified an additional tumor cell subset, other than the nine subsets detected with the pan-tumor panel alone, the merging procedure also generated single-cell data on antibody combinations that are usually not examined together, particularly the expression of certain immune cell markers on tumor and stromal cells. 

With the pan-tumor panel, we were able to identify the main cellular subtypes of the TME. The findings are in accordance with the results from our recent study that describes the establishment of the panel [22]. However, given that analytic tools, samples, and sample sizes differed, the cell populations identified did not overlap completely. While the Astrolabe MATLAB platform identified nine main tumors and one main stromal cell population across the ten patients, a total of four tumors and three stromal cell populations were found when the X-shift algorithm was applied to the three samples in the original publication [22]. Some important considerations when selecting algorithms for analyzing mass cytometry data are the study questions and hypotheses, which parameters could and should be measured, and the output data generated [35]. In our paper from 2021, the focus was on investigating how different tumor tissue dissociation methods influenced the outcome of mass cytometry analysis. In the previous study, one dataset comprising three different patient samples was processed as a single batch for mass cytometry analysis, and the cell subsets, including cell activation markers, were examined across samples for differences that were related to the dissociation method used. The unsupervised X-shift method can be used to detect cell subsets robustly at an in-depth level [35] and was, therefore, applied in the previous study. For the datasets generated in the present research, the FlowSOM and Astrolabe Cytometry Platform algorithms were applied. As the aim was to examine the data generated by the merging method against the two different original datasets, the Astrolabe platform was selected for its automatization function of statistical analyses, as this can decrease the subjectivity of the analysis and increase the reproducibility of the results [30,33]. Astrolabe has previously primarily been used for immunophenotyping of mass cytometry data on single cells isolated from blood samples and, therefore, lacks standardized clustering strategies for tumor or stromal cells. Consequently, we customized the labeling strategies of the analysis pipeline for tumor and stromal cells based on the gating strategy used in the development of the tumor panel [22]. 

Over the last few years, an increasing number of publications have been using mass cytometry to characterize the TME and determine its composition and cell populations at the single-cell level, and such studies have also been reported for EOC and its subtype HGSOC [23,36,37,38,39]. However, most studies describe only specific parts of the TME in detail [36,37,38,39]. We have only identified one publication [25] that, in line with our approach, used a pan-tumor panel that outlines all three main cellular components of the TME. Similar to our panel, the one reported by Casado et al. was also manually designed. Most of the markers included in the two panels established in the present study did not overlap, but they were, still, able to capture the heterogeneous cellular composition of the HGSOC tissue samples and to discriminate between different patient profiles. In contrast, Casado et al. examined sequentially sampled tumors and were, thus, able to detect temporal changes in tumor composition throughout the disease course [25]. Another point of differentiation between the two studies is that the present one used predefined marker-based cell populations in the first analytic step and the unsupervised FlowSOM clustering algorithm for further identifying cell subsets in the second step, while Casado et al. performed customized data analyses in the Cyto solution. Specifically, they first applied the unsupervised clustering method Phenograph to the HGSOC data, before performing further in-depth analyses by running the dimensionality reduction algorithm UMAP on the tumor cell subsets. In this study, the rigidity of the analyses for which the Astrolabe algorithm was implemented was better suited for the dataset comparison; contrastingly, the Cyto solution enables direct investigation of subsets for exploratory studies. Additional exploration of data could be possible by combining the two platforms, for example, by importing the datasets generated in Astrolabe into Cyto for further data examination.

The pan-immune panel used in the present study contains a broader spectrum of lineage markers than those included in the pan-tumor panel. Thus, it was able to provide a more detailed picture of the immune cell composition of the tumors and reveal a new level of immunophenotypic heterogeneity in dissociated tumors from HGSOC patients. The findings demonstrated that despite the considerable differences in the scale of the cell populations detected in the tumors, some cell subsets were present in all the samples and were also detected at relatively high frequencies: double-negative T cells, conventional and type 1 DCs, CD14^+^CD16^−^monocytes/macrophages, NK cells, and two linage-negative cell subsets including a potential tumor-infiltrating macrophage subset. The high proportion of DCs and monocyte/macrophage subsets detected is in agreement with the results from other EOC studies [40,41,42,43]. This implies that myeloid cells, particularly tumor-associated macrophages and DCs, could be important modulators of T-cell responses in ovarian cancer. These results indicate that a combined analysis of myeloid and lymphoid lineage cells, as well as tumor cells, may be required to determine whether a patient is suitable for immunotherapeutic modalities such as DC-based vaccines and CAR NK cell therapy. 

Increasingly advanced technologies and multiplexed analyses are currently being applied for cellular profiling, for example, fluorescence-based methods such as flow cytometry and immunofluorescence imaging, multiplex immunohistochemistry, mass cytometry, protein and cytokine panel analysis, T- and B-cell receptor sequencing, and RNA and/or DNA sequencing [44,45]. This development has occurred in parallel with the realization that insights into the span of single cells and their phenotypic and functional variability require the development of techniques for single-cell analysis as opposed to bulk analysis. While single-cell proteomics methods, such as mass cytometry, can help trace trajectories and rapid and dynamic changes in the TME caused by disease progression or therapies, genetics-based methods contribute to more static TME profiling [46,47,48]. Several studies have applied multiplexing methods to examine the EOC TiME and have reported differing patterns of tumor-infiltrating lymphocytes, as well as associations between CD47 expression on different immune cells and survival parameters, and associations between the distribution of immune cell phenotypes and survival or response to combined therapy with poly-ADP-ribose polymerase and immune checkpoint inhibitors [44,45,49,50]. The present study has also illustrated the identification of a wide array of single-cell phenotypes by mass cytometry-based profiling of the TME, with at least the same level of complexity as other methods; however, the sample size in the present study made the detected clinical associations less reproducible than those reported by Färkkilä et al. [44]. 

Several studies on solid tumors have demonstrated the value of applying multiple markers for a more detailed characterization of cellular phenotypes at the single-cell level [51]. The dataset (batch) combining approach facilitates analytic flexibility and improved statistical capacity, despite challenges with inter-batch variation in data structure or the defining of phenotypic markers of cell subsets, as well as the controversies associated with data imputation [52]. The use of the CyTOFmerge algorithm enabled both the extension of the number of markers per cell and the combination of overlapping markers from the existing mass cytometry datasets to create a shared marker set between the two panels. This allowed for a deeper interrogation of the cellular composition of the HGSOC TME, as well as the characterization of specific cell subtypes in detail, as it provided a better picture of the heterogeneity of the data. The method enabled the characterization of the abundant cell populations in more detail and the identification of cell populations expressing certain antibody combinations that could be explored further. CyTOFmerge is a computational method for the integration of measurements from multiple mass cytometry panels. In the original paper, Abdelaal et al. were able to show how data imputation, which has previously mostly been applied in genomic analyses, can be applied to cytometry data as well [27]. The selected set of shared information captured the underlying structure of the data. On the other hand, this study found, in line with Abdelaal et al., that marker imputation influences the value of the downstream validation when matched with non-imputed data, and also when the pipeline used followed the step-wise approach suggested.

The pan-tumor panel identified nine tumor cell subsets, which were defined as being negative for CD45 and CD56, but expressing at least one of the three markers, namely, EpCAM, FOLR1, and CD24. The merging of the datasets led to the detection of one additional tumor cell subset that expressed markers traditionally examined in immune cells. In this study, cross-imputation of data based on overlapping panel markers resulted in similar, but not completely identical, marker expression on the common cell subsets. This contrasts somewhat with the findings of Pedersen et al., who demonstrated replication of the same cell subsets with identical expression levels after merging panel data by using the merging algorithm included in cyCombine [53]. These differences could be partially explained by discrepancies in the merged panels and in the sample origin and size. For example, while Pedersen et al. merged two immune cell-based panels obtained using blood samples from healthy donors and patients with chronic lymphocytic leukemia, the present study focused on immune cells and a broader TME panel of dissociated tumors [53]. Still, the discrepancies between marker expression on subsets found in the merged data versus that identified in the pan-tumor and pan-immune data illustrate a weakness of the method. This disappearance was also marked in the discordant results from the analysis of phenotype-associations of the merged data compared to the results from the other two datasets (Table 2). The finding underscores potential issues in the CyTOFmerge process to be aware of and supports the requirement for validation of findings by other methods.

The combining of several mass cytometry datasets for better exploration of data is increasingly being attempted and is supported by the development of more merging algorithms [53,54,55]. One of the emerging algorithms is cyCombine, which, similar to CyTOFmerge, is based on the markers that overlap between the examined panels and the imputation of the remaining markers [27,53]. Contrary to earlier methods for merging that involved only the imputation of data from one dataset to another [56], both CyTOFmerge and cyCombine allow for the imputation of data into all the included datasets. While CyTOFmerge applies a *k*-nearest-neighbor algorithm for imputation, this process is performed through weighting-based self-organizing mapping in cyCombine. Furthermore, cyCombine includes batch correction prior to the merging of data, and this lowers the risk of false-positive findings in discrepant datasets. Apparently, the added value of this newer data analysis tool, introduced in 2022, compared to CyTOFmerge, is the inclusion of a batch correction module that reduces the variation in data structure between the analyzed datasets and, thereby, increases the robustness of findings. In our study, the samples were prepared in the same lab on the same day, and the same samples and antibodies (the same panel antibody mixture for all samples as well as the same batch for those antibodies that overlapped between the two panels) were used. Further, data was barcoded and collected from the Helios machine on the same day to minimize errors. The lack of batch correction with CyTOFmerge will be considered in further merging investigations. With the increasing use of multiomics approaches to identify molecular targets and therapeutic strategies in cancer, there is an increased need for advanced analytic tools for the combination of multiplexed data. This study is the first to apply a merging algorithm to solid tumors from HGSOC patients, and the results imply that such merging of data could add value from a hypothesis-generating perspective and open up novel avenues for ovarian cancer research.

The limitations of the current study include the small sample size and the inclusion of single tumor site biopsy samples, which restricted the interpretation of the findings, particularly with regard to the suggested clinical associations. However, the sample size did not have a pronounced effect on the exploration of the merging algorithm, as the same samples were examined by two mass cytometry panels for the analyses. Another limitation was that further investigations of specific cell subsets, such as monocytes/macrophages, were restricted by the lack of complementary markers in the panel. The few immune markers included in the pan-tumor panel enabled only the identification of the main subsets of leukocytes in the resulting dataset. Furthermore, all CD45^−^cells were excluded from further analyses of the pan-immune data after the Astrolabe analyses. Therefore, a comparison of tumor marker expression on immune subsets was made impossible between the three datasets, including the imputed markers in the merged data. Hence, dedicated panels are required for more in-depth profiling of specific cell subsets, as well as cells with new marker combinations identified by the merging algorithm. The use of single-cell suspensions for the profiling does not allow us to map spatial information on TME histological patterns. In the future, another layer of TME profiling can be generated through multiplexed immunofluorescence analyses or through imaging mass cytometry of tissue sections [57,58,59]. To examine new combinations of antigens expressed on stromal and tumor cells, and to establish associations between cell subsets and clinical outcomes, further exploration in larger HGSOC cohorts with multiple-site biopsies is required, preferentially in combination with other methods, such as single-cell RNA sequencing, spatially resolved transcriptomics, and imaging mass cytometry.

## 5. Conclusions

To improve therapeutic options and treatment specificity, the heterogeneity of the HGSOC TME needs to be better mapped. In this study, we demonstrate how the merging of datasets from the same cohort could be beneficial for the identification of new TME cell subsets of interest and cells expressing antigen combinations that are unlikely to be examined together otherwise. Combining complementary data in this way may have hypothesis-generating potential and help in the identification of potential new cell subsets that can be investigated in depth by customized panels and supplementary methods in larger cohorts and lead to the generation of new knowledge in the field of tumor pathophysiology. 

## Figures and Tables

**Figure 1 cancers-15-05106-f001:**
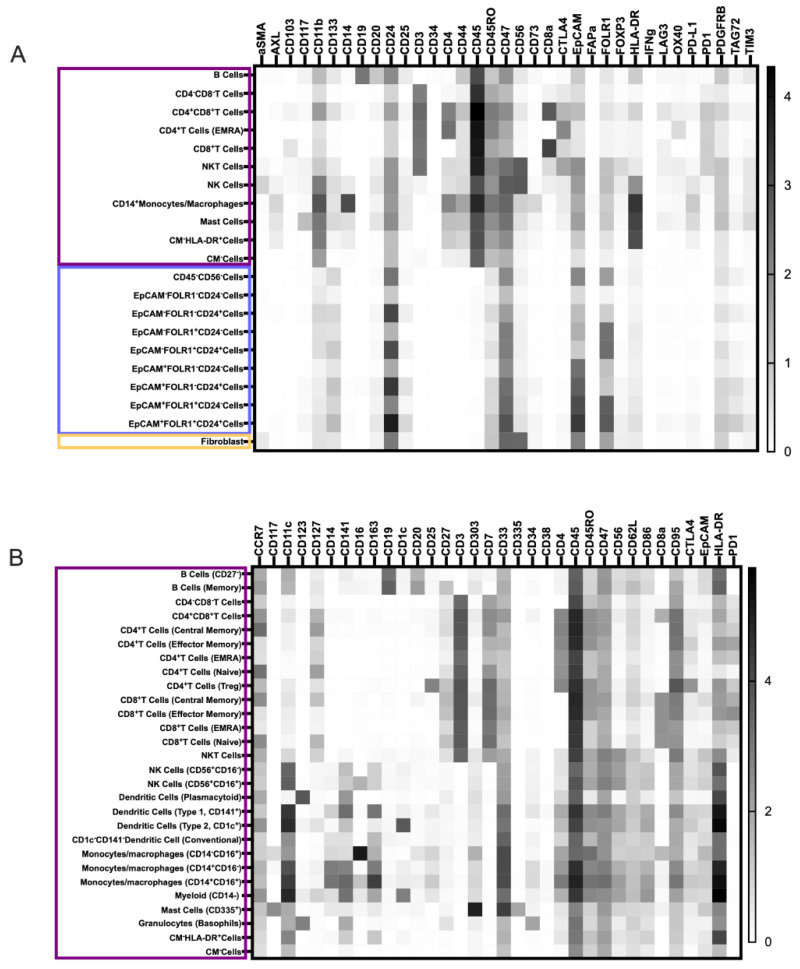
Heatmaps illustrating the median expression of antibodies (horizontally) in the cell populations pre-defined by the software algorithm. (**A**) The pan-tumor panel data, (**B**) the pan-immune panel data, and (**C**) the merged dataset. The purple boxes indicate immune cell populations; the blue boxes, tumor cell populations and the yellow boxes, fibroblast populations.

**Figure 2 cancers-15-05106-f002:**
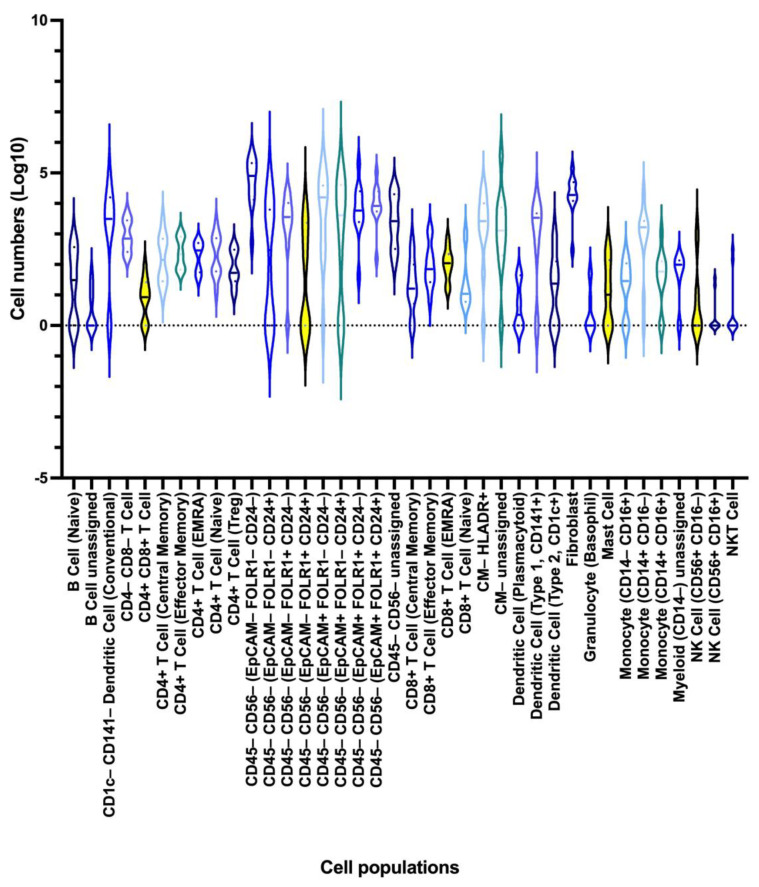
Violin plot depicting interpatient differences in cell populations (n = 39) based on the merged dataset. The cell populations are presented along the X-axis, and the log10-transformed cell numbers are displayed along the Y-axis. The horizontal lines in the violins indicate the median cell number of that population, while the dots indicate the 1st and 3rd quartiles.

**Figure 3 cancers-15-05106-f003:**
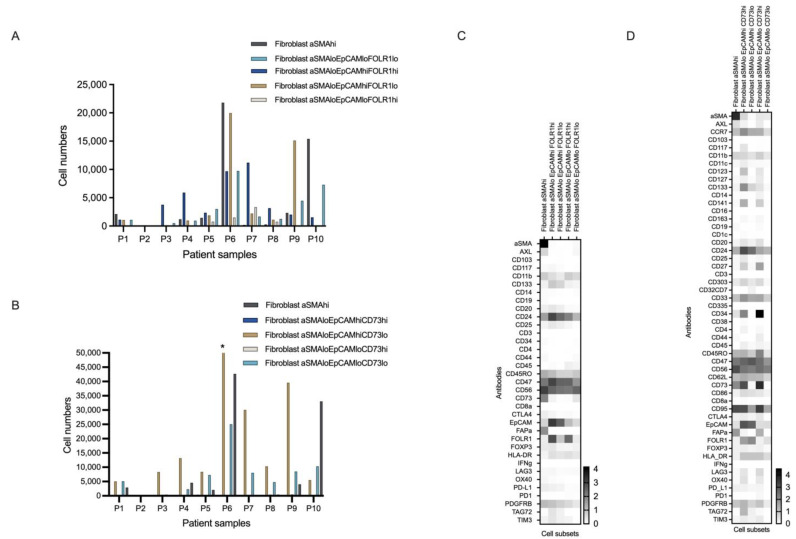
Cell populations and antibody expression intensity of the five identified fibroblast subsets. (**A**,**B**) The bar charts illustrate heterogeneity in the five fibroblast subset populations across the 10 patient samples determined with the pan-tumor panel (**A**) and the merged panel (**B**). The X axes present the samples, while the Y axes present the cell numbers. The asterisk indicates an outlying value (>77,000 cells) for the number of aSMA^lo^EpCAM^hi^CD73^lo^ fibroblasts in sample P6. (**C**,**D**) Heatmaps of antibody expression in the five fibroblast subsets defined by the clustering algorithm FlowSOM in the pan-tumor panel (**C**) and the merged marker dataset (**D**). Antibodies are shown on the Y axis, while the five cell subsets are placed along the X axis.

**Figure 4 cancers-15-05106-f004:**
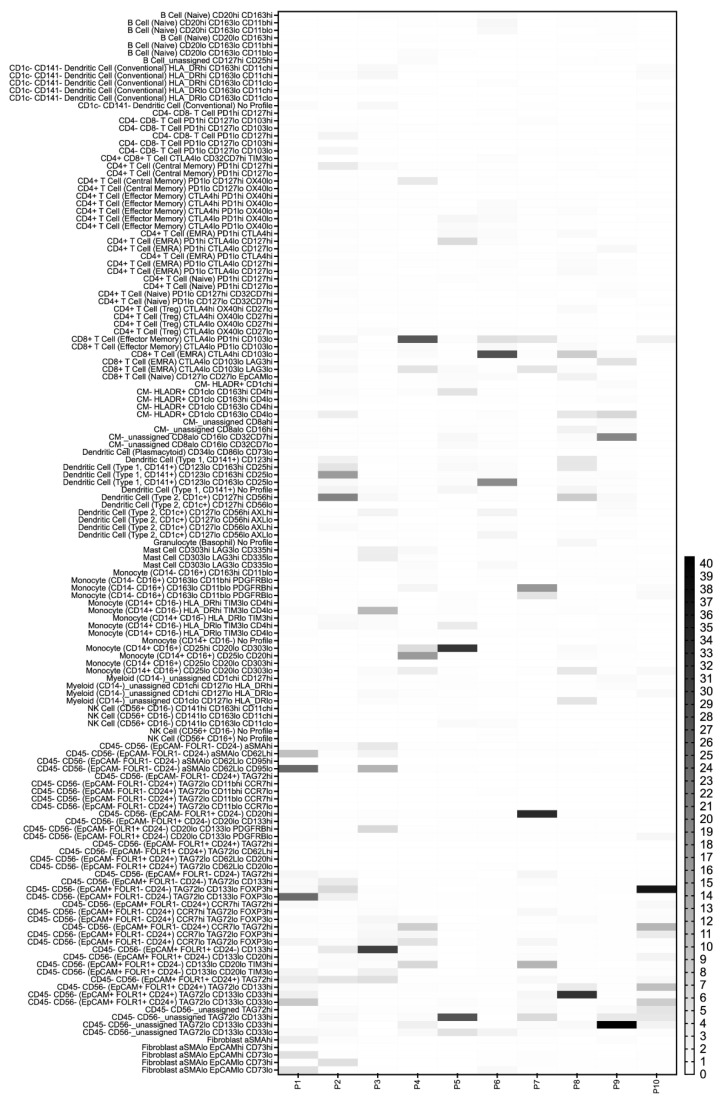
Heatmap depicting the fractions of the most common cell subsets in each sample. The cell subsets are presented on the vertical axis, and the patients are presented along the horizontal axis. The percentage of a subset in each sample is depicted, with dark colors indicating higher frequencies. Only cell subsets containing ≥50 cells are presented for the ten patient samples.

**Table 1 cancers-15-05106-t001:** Characteristics of the studied cohort of patients with HGSOC (N = 10).

Phenotype		Numbers
Age		
	Median years (range)	70 (54–78)
Stage of cancer		
	I–II, III–IV	4:6
Surgical outcomes		
	R < 1, R > 1	7:3
Clinical status at time of analysis		
	Disease-free, living with disease, dead	6:2:2

Except for age, for which the median and range values are provided, the other parameters are expressed as the number of patients.

**Table 2 cancers-15-05106-t002:** Associations between phenotypes and pre-defined cell populations (n = 39).

Dataset	Phenotype	Cell Subset	logFC	*p* Value	FDR
Pan-tumor					
	Recurrence	B cells	−4.41	0.006	0.13
Pan-immune					
	Clinical status	CD4^+^CD8^+^T cells	−5.94	0.0003	0.01
		Plasmacytoid dendritic cells	−11.49	0.009	0.13
Merged					
	Recurrence	CD56^+^CD16^−^NK cells	−13.76	0.0018	0.07
		NKT cells	−10.46	0.008	0.16
	Clinical status	EpCAM^−^FOLR1^+^CD24^+^cells	−14.64	0.0003	0.01
		Plasmacytoid dendritic cells	−12.32	0.007	0.07
		CD56^+^CD16^−^NK cells	−13.50	0.007	0.07
		Unassigned cells	−10.18	0.008	0.07
		Unassigned HLA-DR^+^cells	−11.49	0.009	0.07

logFC; logarithmically transformed fold change, FDR; false discovery rate, NK; natural killer, EpCAM; epithelial cellular adhesion molecule, FOLR1; folate receptor alpha, HLA-DR; major histocompatibility complex (MHC) II cell surface receptor.

## Data Availability

The data presented in this study are available on request from the corresponding author. The data are not publicly available due to due to ethical restrictions.

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
