# Peer review of "Combining Mass Cytometry Data by CyTOFmerge Reveals Additional Cell Phenotypes in the Heterogeneous Ovarian Cancer Tumor Microenvironment: A Pilot Study"

_cancers, 2023, doi:10.3390/cancers15205106_

Round 1
Reviewer 1 Report
Thomsen et al presented the pilot study concerning to the combined use of mass cytometry data in TME of ovarian cancer. This article is a good piece of work, data well presented, and concluded suitably, hence, it deserves publication.
Author Response
Reviewer 1
1) Thomsen et al presented the pilot study concerning to the combined use of mass cytometry data in TME of ovarian cancer. This article is a good piece of work, data well presented, and concluded suitably, hence, it deserves publication.
Reply:
We are grateful that Reviewer 1 finds our study of value and the manuscript worthy of publication.
Reviewer 1
1) Thomsen et al presented the pilot study concerning to the combined use of mass cytometry data in TME of ovarian cancer. This article is a good piece of work, data well presented, and concluded suitably, hence, it deserves publication.
Reply:
We are grateful that Reviewer 1 finds our study of value and the manuscript worthy of publication.
Reviewer 2 Report
In this manuscript the authors used an existing algorithm CyTOFmerge to integrate cells from the same patient cohort with either 35 pan-tumor markers or 34 pan-immune markers and identify a main tumor cell subset additionally to the nine identified by the pan-tumor panel. Also, several main cell types are found to be associated with either the recurrence or clinical status. Nevertheless, there are substantial concerns regarding both the methodology and the outcomes. Primarily, the reliability of CyTOFmerge for single-cell CyTOF integration is contingent not solely on the shared number of protein markers, but also on the extent of overlap between the cell types represented in the two panels. This distinction sets it apart from bulk tumor integration, where only the number of overlapping markers is critical. In this instance, despite having 18 protein markers common to both panels, the immune panel excluded CD45- cells from subsequent analysis. Consequently, the integration of the immune panel does not influence the tumor/stroma clusters within the tumor panel. Failure to acknowledge this distinction may lead to unusual results in Section 3.4.1, where the manuscript discusses the emergence of so-called 'novel cell subsets' post-integration, falsely attributing cancer markers to immune cells. Furthermore, even when examining immune cell subsets, caution must be exercised when utilizing CyTOFmerge. This is because each immune cell type within the tumor panel typically expresses fewer than five markers, excluding CD45 and CD45RO, as indicated in Figure 1A. It is not reasonable to impute additional markers from such limited data, especially when these markers may not be shared between the two panels, and these proteins are typically chosen as distinct markers for independent cell types. Finally, the peculiar disappearance of phenotype-associated subsets identified by each panel after integration, as observed in Table 2, underscores potential issues in the CyTOFmerge process.
Minor issues include:
1. In line 201, it is mentioned that "a panel containing 35 antibodies was used." However, the count reaches 36 antibodies when considering tumor cells (n = 11), stromal cells (n = 4), immune cells (n = 15), and immune checkpoint markers (n = 6). This raises questions about accuracy. Is there an overlap between immune cell markers and immune checkpoint markers? Additionally, only 25 antibodies have a clear source, either from the manufacturer (Fluidigm) (n = 14) or in-house (n = 11). What about the remaining 10 antibodies?
2. In lines 258-261, it is stated that UMAP is utilized in supplementary figures S2 and S10. However, S2 employs viSNE instead of UMAP. Moreover, the arrangement of PBMC and OC90 in S2 appears incorrect, and the subplot figures in both S2 and S10 are too small.
3. The description of how clustering and annotation are conducted for the cells is convoluted. In line 273, it is mentioned that data from both panels are clustered using Phenograph. Yet, in the following paragraph (lines 276-278), it is stated that the Astrolabe platform is used to label each sample individually based on the Ek'Balam algorithm of pre-defined cell subsets. Subsequently, the next line mentions, "Further data clustering was performed using the consensus hierarchical clustering tool FlowSOM." In lines 302-304, it is noted, "Both the pre-defined immune cell populations and the cell subsets identified by the unsupervised clustering procedure were used to determine potential clinically relevant differences between samples." This section introduces at least three different clustering methods, and it remains unclear which one is applied to which cells, particularly in the context of unsupervised clustering.
Author Response
1) In this manuscript the authors used an existing algorithm CyTOFmerge to integrate cells from the same patient cohort with either 35 pan-tumor markers or 34 pan-immune markers and identify a main tumor cell subset additionally to the nine identified by the pan-tumor panel. Also, several main cell types are found to be associated with either the recurrence or clinical status. Nevertheless, there are substantial concerns regarding both the methodology and the outcomes.
Primarily, the reliability of CyTOFmerge for single-cell CyTOF integration is contingent not solely on the shared number of protein markers, but also on the extent of overlap between the cell types represented in the two panels. This distinction sets it apart from bulk tumor integration, where only the number of overlapping markers is critical. In this instance, despite having 18 protein markers common to both panels, the immune panel excluded CD45- cells from subsequent analysis. Consequently, the integration of the immune panel does not influence the tumor/stroma clusters within the tumor panel. Failure to acknowledge this distinction may lead to unusual results in Section 3.4.1, where the manuscript discusses the emergence of so-called 'novel cell subsets' post-integration, falsely attributing cancer markers to immune cells. Furthermore, even when examining immune cell subsets, caution must be exercised when utilizing CyTOFmerge. This is because each immune cell type within the tumor panel typically expresses fewer than five markers, excluding CD45 and CD45RO, as indicated in Figure 1A. It is not reasonable to impute additional markers from such limited data, especially when these markers may not be shared between the two panels, and these proteins are typically chosen as distinct markers for independent cell types.
Reply:
We appreciate the thoroughness of the review. The Reviewer has a valid point, and we have tried to address this in the revised version of the manuscript by changing the paragraph 3.4.1 Novel Cell Subsets. The merging of the pan-tumor and the pan-immune datasets, and thereby the imputation of immune markers from the pan-immune dataset onto tumor cell subsets in the merged data, was performed prior to the removal of CD45-cells in the Astrolabe algorithm. Therefore, it is reasonable to inspect the markers from the pan-immune dataset imputed on CD45-cells in the merged data. As the Reviewer indicates, the immune markers in the pan-tumor panel are few, and by use of these data only the main immune cell subsets can be identified. In the revised text we have removed the text describing results related to the immune populations with imputed tumor cell marker expression. We have also added a sentence to the paragraph of the discussion, where limitations are deliberated on:
The relevant part of the text, found in paragraph 3.4.1 Novel Cell Subsets (p.21), now reads:
"The limitations of the current study include the small sample size and the inclusion of single tumor site biopsy samples, which restricted the interpretation of the findings, particularly with regards to the suggested clinical associations. However, the sample size did not have a pronounced effect on the exploration of the merging algorithm, as the same samples were examined by two mass cytometry panels for the analyses. Another limitation was that further investigations of specific cell subsets, such as monocytes/macrophages, was restricted by the lack of complementary markers in the panel. The few immune markers included in the pan-tumor panel enabled only identification of the main subsets of leukocytes in the resulting dataset. Furthermore, all CD45-cells were excluded from further analyses of the pan-immune data after the Astrolabe analyses. Therefore, comparison of tumor marker expression on immune subsets was made impossible between the three datasets, including the imputed markers in the merged data. Hence, dedicated panels are required for more in-depth profiling of specific cell subsets, as well as cells with new marker combinations identified by the merging algorithm."
2) Finally, the peculiar disappearance of phenotype-associated subsets identified by each panel after integration, as observed in Table 2, underscores potential issues in the CyTOFmerge process.
Reply:
We appreciate the observation, and we have changed the text of the Discussion part (p.20-21) accordingly:
"The pan-tumor panel identified nine tumor cell subsets which were defined as being negative for CD45 and CD56, but expressing at least one of the three markers EpCAM, FOLR1, and CD24. The merging of the datasets led to the detection of one additional tumor cell subset that expressed markers traditionally examined in immune cells. In this study, cross-imputation of data based on overlapping panel markers resulted in similar, but not completely identical, marker expression on the common cell subsets. This contrasts somewhat with the findings of Pedersen et al., who demonstrated replication of the same cell subsets with identical expression levels after merging panel data by using the merging algorithm included in cyCombine (54). These differences could be partially explained by discrepancies in the merged panels and in the sample origin and size. For example, while Pedersen et al. merged two immune cell-based panels obtained using blood samples from healthy donors and patients with chronic lymphocytic leukemia, the present study focused on immune cells and a broader TME panel of dissociated tumors (54). Still, the discrepancies between marker expression on subsets found in the merged data versus that identified in the pan-tumor and pan-immune data illustrates a weakness of the method. This disappearance was also marked in the discordant results from the analysis of phenotype-associations of the merged data compared to the results from the other two datasets (Table 2). The finding underscores potential issues in the CyTOFmerge process to be aware of and supports the requirement for validation of findings by other methods."
Minor issues include:
- In line 201, it is mentioned that "a panel containing 35 antibodies was used." However, the count reaches 36 antibodies when considering tumor cells (n = 11), stromal cells (n = 4), immune cells (n = 15), and immune checkpoint markers (n = 6). This raises questions about accuracy. Is there an overlap between immune cell markers and immune checkpoint markers?
Additionally, only 25 antibodies have a clear source, either from the manufacturer (Fluidigm) (n = 14) or in-house (n = 11). What about the remaining 10 antibodies?
Reply:
The numbers of markers annotating the three main cell types of the TME do indeed add up to 36. As the Reviewer points out, one marker is included in two of the groups, the stromal cells and the immune cells: CD56, as this is both a marker for fibroblasts for a subset of natural killer cells.
Thankfully, the Reviewer point out a spelling mistake with regards to the number of antibodies. The number of antibodies sourced from Fluidigm was n=24, not n=14. We appreciate the opportunity to correct this number.
- In lines 258-261, it is stated that UMAP is utilized in supplementary figures S2 and S10. However, S2 employs viSNE instead of UMAP. Moreover, the arrangement of PBMC and OC90 in S2 appears incorrect, and the subplot figures in both S2 and S10 are too small.
Reply:
The analysis applied is UMAP, both in S2 and in S10, as can be seen marked UMAP_1/UMAP_2 for each UMAP displayed in the figures. Unfortunately, we have written in the figure legend of Figure S2 that we have employed viSNE, although this is incorrect. In the revised version of the Supplementary material, we have corrected the figure legend accordingly.
The arrangement of the UMAP illustrating the PBMCs for the pan-immune data has been moved down to align with the UMAPs of PBMCs generated from the pan-tumor and merged data. We agree that the subplot figures are small in the two supplementary figures S2 and S10 and have enlarged the figures accordingly.
- The description of how clustering and annotation are conducted for the cells is convoluted. In line 273, it is mentioned that data from both panels are clustered using Phenograph. Yet, in the following paragraph (lines 276-278), it is stated that the Astrolabe platform is used to label each sample individually based on the Ek'Balam algorithm of pre-defined cell subsets. Subsequently, the next line mentions, "Further data clustering was performed using the consensus hierarchical clustering tool FlowSOM." In lines 302-304, it is noted, "Both the pre-defined immune cell populations and the cell subsets identified by the unsupervised clustering procedure were used to determine potential clinically relevant differences between samples." This section introduces at least three different clustering methods, and it remains unclear which one is applied to which cells, particularly in the context of unsupervised clustering.
Reply:
It is correct that use of both Phenograph and FlowSOM is described in the text. The CyTOFmerge applies Phenograph in the last part of the merging algorithm, while the Astrolabe platform applies FlowSOM as the clustering algorithm in the second step of the analysis, after the pre-determined cell subsets were identified through the first step of the analysis pipeline. We understand that we have been imprecise in our descriptions. The text has been revised in the new version of the manuscript (lines 273, 305, and 306).
Reviewer 3 Report
This manuscript is well written, and experiments were well executed. In this article authors conducted extensive analysis of 10 HGSOC patient tumors using mass cytometry. Authors utilized 2 different panels to characterize tumor microenvironment and tumor immune microenvironment. Additionally, they combined both panel datasets using CyTOFmerge MATLAB software and additional tumor microenvironment cell phenotypes were discovered. With mass cytometry we will not be able to capture all cell phenotypes because of limited number of markers. While authors tried to combine 2 different datasets information is still not complete. I agree with authors that combining mass cytometry with single cell RNA sequencing and methods like this will solve this problem. I believe this manuscript will generate more interest to readers if authors could elaborate on frequencies of cell populations and their clinical associations such as therapy resistance, metastasis, and relapse.
Author Response
1) This manuscript is well written, and experiments were well executed. In this article authors conducted extensive analysis of 10 HGSOC patient tumors using mass cytometry. Authors utilized 2 different panels to characterize tumor microenvironment and tumor immune microenvironment. Additionally, they combined both panel datasets using CyTOFmerge MATLAB software and additional tumor microenvironment cell phenotypes were discovered.
With mass cytometry we will not be able to capture all cell phenotypes because of limited number of markers. While authors tried to combine 2 different datasets information is still not complete. I agree with authors that combining mass cytometry with single cell RNA sequencing and methods like this will solve this problem.
Reply:
We appreciate both the comments on the manuscript and the study execution, and that the Reviewer agrees with our suggestions for improving the phenotype examination by combining the methods mass cytometry and single cell RNA in further experiments.
2) I believe this manuscript will generate more interest to readers if authors could elaborate on frequencies of cell populations and their clinical associations such as therapy resistance, metastasis, and relapse.
Reply
The additional analyses suggested by the Reviewer regarding clinical associations with the different cell populations and how frequently these are in the tumor samples would be truly interesting. Unfortunately, the sample size of this pilot study does not allow for fine-grained investigations, as the numbers of patients with the each of the clinical outcomes would be too small to reach reproducible results. These investigations clinically relevant analyses should be part of the analysis plan of a larger follow-up study.